# Study on Characteristics and Lignification Mechanism of Postharvest Banana Fruit during Chilling Injury

**DOI:** 10.3390/foods12051097

**Published:** 2023-03-04

**Authors:** Lu Xiao, Xunyuan Jiang, Yicai Deng, Kaihang Xu, Xuewu Duan, Kai Wan, Xuemei Tang

**Affiliations:** 1Institute of Quality Standard and Monitoring Technology for Agro-Products of Guangdong Academy of Agricultural Sciences, Guangzhou 510640, China; 2Key Laboratory of Testing and Evaluation for Agro-Product Safety and Quality, Ministry of Agriculture and Rural Affairs, Guangzhou 510640, China; 3Guangdong Provincial Key Laboratory of Quality & Safety Risk Assessment for Agro-Products, Guangzhou 510640, China; 4Guangdong Provincial Key Laboratory of Applied Botany, South China Botanical Garden, Chinese Academy of Sciences, Guangzhou 510650, China

**Keywords:** banana peel, chilling symptoms, cell wall metabolism, lignification mechanism

## Abstract

The banana is prone to chilling injury (CI) at low temperature and showing a series of chilling symptoms, such as peel browning, etc. Lignification is a response to abiotic stress and senescence, which is an important manifestation of fruits and vegetables during chilling exposure. However, little is known about the lignification of bananas during low-temperature storage. Our study explored the characteristics and lignification mechanism of banana fruits during low-temperature storage by analyzing the changes of chilling symptoms, oxidative stress, cell wall metabolism, microstructures, and gene expression related to lignification. The results showed that CI inhibited post-ripening by effecting the degradation of the cell wall and starch and accelerated senescence by increasing O^2−^ and H_2_O_2_ content. For lignification, *Phenylalanine ammonia-lyase* (*PAL*) might start the phenylpropanoid pathway of lignin synthesis. *Cinnamoyl-CoA reductase 4* (*CCR4*), *cinnamyl alcohol dehydrogenase 2* (*CAD2*), and *4-coumarate--CoA ligase like 7* (*4CL7*) were up-regulated to promote the lignin monomer’s synthesis. *Peroxidase 1* (*POD1*) and *Laccase 3* (*LAC3*) were up-regulated to promote the oxidative polymerization of lignin monomers. These results suggest that changes of the cell wall structure and cell wall metabolism, as well as lignification, are involved in the senescence and quality deterioration of the banana after chilling injury.

## 1. Introduction

Low-temperature storage, as one of the most useful technologies, can slow down the speed of cell metabolism and delay plant senescence to extend the post-harvest life of horticultural products [1,2]. However, some tropical or subtropical fruits, such as banana and mango, are highly susceptible to chilling injury (CI). Low-temperature storage would cause appearance of physiological disorders and then negatively affect the fruit’s quality and, therefore, their marketing [3,4,5]. The CI symptoms can be observed in all developmental stages of the plant, which is related to the storage time, temperature of cold treatment, plant organ, and even developmental stage [1]. The symptoms of CI include skin depression, abnormal skin yellowing, tissue decomposition, internal or surface browning, water infiltration in intercellular spaces, the development of a woolly or dry texture in the tissue, as well as developmental or metabolic disorders, such as incomplete ripening and deficient flavor and aroma. The injured tissue or organ expressed a lower resistance to mechanical injury, fungal infection, and microbial infection [6]. Some studies suggested that the texture changes of certain CI-sensitive fruits during low-temperature storage are due mainly to disorders of cell wall metabolism, including a decrease in the solubalization and depolymerization of pectins [7,8,9,10].

Lignification, a response to abiotic stress and natural senescence, is observed in many chilling-sensitive horticultural products, such as loquat [11], mangosteen [12], bamboo shoots [13], and pears [14]. Generally, lignification is caused by the accumulation of lignin in plant cell walls and the biosynthesis of lignin has been widely studied in the model plant Arabidopsis, woody tress, and some other crops [15,16]. However, the process of lignification has different pathways, and the synthesis and polymerization of lignification varies among different plant species and cultivars [17]. Banana, as an typical tropical or subtropical fruit, is sensitive to chilling and suffers physiological damage when exposed to low-temperature (<13 °C) storage, generally resulting in peel browning, pitting, discoloration, and abnormal fruit ripening and softening [18,19,20,21]. Chilling injury in horticultural products has been attributed to alterations in cell membrane structure and oxidative stress [1]. Disorders in the composition of lipids, unsaturation, and the phase transition were accepted to be the primary response to CI, which lead to the changes of cell membranes components. Although the evident differences in CI symptoms have been studied, little attention has been paid to CI-induced lignification in the banana fruit.

In this study, the objective was to investigate the characteristics of CI as well as the mechanism of lignification induced in harvested banana fruits during storage at a low temperature. Some related physiological indexes, including chilling symptoms, cell wall component, enzyme activity, observation of tissue structure, and related gene expression, were tested and analyzed. This study provides a theoretical basis for the study of alleviating the symptoms of banana chilling injury by exploring its mechanism.

## 2. Materials and Methods

### 2.1. Plant Materials and Treatments

Green mature banana (*Musa* spp., AAA group cultivar “Brazil″) fruit (approximately 110 days after anthesis) was harvested from an orchard in Shawei Village, Wanqingsha Town, Nansha District, Guangzhou city, Guangdong province, China (E 113°35′20.93″, N 22°39′35.68″) and transported to the laboratory within 2 h. Banana fruit fingers were cut and then dipped in 0.1% Sportak^®^ (Prochloraz, Bayer, Germany) fungicide solution for 3 min and finally allowed to air-dry. Fruit with uniform shape, color, and size were selected and randomly divided into two groups, which were stored at 25 °C or 6 °C respectively with 85–90% relative humidity depending on the allocated group. During storage, samples were periodically taken to measure the related physiological indexes. Some indicators related to the appearance, senescence, cell wall metabolism were measured. Some enzymes were related to the phenylpropanoid pathway and peel browning, including the up-stream enzymes PAL, PPO, and POD, were analyzed. Banana peel tissues were collected, frozen in liquid nitrogen, and stored at −80 °C for RNA extraction and physicochemical indexes analysis. Each treatment consisted of three biological replicates, with each replicate containing 60 fruit fingers.

### 2.2. Physiological Parameters

Firmness. Fruit firmness was measured with a penetrometer (Model FT 327, Facchini, Alfonsine, Italy) by recording the force needed to penetrate 5 mm into the middle part of whole fruit. The flat probe of penetrometer is 8 mm in diameter. The results were expressed as N.

Peel color. Banana peel color was measured with a Konica Minolta CR-400 colorimeter (Konica Minolta Co. Ltd., Japan) in the CIEL*a*b mode. L* indicated the lightness ranging from black to white, C* represented the color saturation that varied from dull to vivid, and hue angle (h*) referred to a color wheel: red at an angle of 0°, yellow at 90°, green at 180°, and blue at 270° in accordance with the method by McGuire [22].

Chlorophyll fluorescence. Chlorophyll fluorescence of the banana peel was determined using a portable chlorophyll fluorometer (PAM 2100, Walz, Germany). Minimum and maximum fluorescence yields (*Fo, Fm*) were measured on dark-adapted (30 min) fruit. *Fo* was measured with a measuring beam at a light intensity less than 0.05 μmol m^−^^2^ s^−^^1^, while Fm was obtained by measuring chlorophyll fluorescence during a 2.5 s pulse of saturating light (18,000 μmol m^−^^2^ s^−^^1^). The maximal variable fluorescence (*Fv* = *Fm*-*Fo*) and PSII quantum yield (*Fv/Fm*) were calculated from the *Fo* and *Fm* values.

Protopectin content. Protopectin was extracted and determined according to the method of Lu et al. [23] with some modifications. Frozen banana peel (5.0 g) was extracted in 100.0 mL 95% ethanol with boiling for 30 min. After cooling to room temperature, the tissue was centrifuged (5000× *g*) for 10 min at 25 °C and the supernatant was removed. Then, the residue was washed twice with 10.0 mL ethanol and the supernate was removed. The precipitate was hydrolyzed in H_2_SO_4_ (0.5 M) at 90 °C for 1 h. After cooling to room temperature, the sample was centrifuged (8000× *g*) for 15 min at 25 °C and the supernate was collected for protopectin analyzation. The following experimental operations mainly referred to corresponding assay kits (Comin biotechnology Co., Ltd., Suzhou, China) in accordance with the manufacturer’s instructions. The absorbance was analyzed by Multiskan Spectrum (Multiskan Go 1510, Thermo Fisher Scientific Oy Ratastie 2, Vantaa Finland) at 530 nm. Protopectin content (mg g^−^^1^ FW) = m_st_ * V_TS_ * ΔA2/(ΔA1 * m * V_S_); ‘m_st_’—the quality of the standard substance, mg; ‘V_TS_’—total volume of enzyme-extracting solution, mL; ‘V_S_’—volume of the reactive enzyme liquid added; ‘m’—fresh weight of banana peel, g; ΔA1_530nm_ = A_standard_ − A_blank_, ΔA2_530nm_ = A_test_ − A_control_.

Cellulose content. Cellulose contents were determined using the anthrone method. Frozen banana peel (3.0 g) was homogenized in 10.0 mL ethanol of 80% for 20 min at 90 °C. After cooling to room temperature, the homogenate was centrifuged (6000× *g*) for 10 min at 25 °C and the supernate was removed. The precipitate was added to 1.0 mL DMSO for 15 h to remove starch and then centrifuged (6000× *g*) at 25 °C for 10 min to remove the supernate. The precipitate was dried as cell wall materials (CWM). A total of 50.0 mg of dried CWM was homogenized in distilled water (5.0 mL) and then added 7.5 mL H_2_SO_4_ slowly and incubated in ice water for 30 min. Then, the homogenate was centrifuged at 4 °C for 10 min (8000× *g*) and the supernate was collected for cellulose content analysis. After adding anthrone reagent and incubating at 95 °C for 10 min, the absorbance of sample was measured against a distilled water blank at 620 nm. Cellulose content (mg^−^^1^ g^−^^1^ DW) = [(ΔA_620nm_ + 0.0043) * V_S/_5.25]/(m * V_S/_Vt); ‘ΔA_620nm_’ = A_test_ − A_blank_; ‘V_S_’—volume of the extracting solution added, mL; ‘V_T_’—total volume of extracting solution, mL; ‘m’—dry weight of banana peel, g; ‘20’—dilution ratio of sample; ‘0.0043’ and ‘5.25’ were from the regression equation (y = 5.25x − 0.0043, R^2^ = 0.9987) under standard conditions.

Lignin content. Lignin was extracted and measured by the method of Bruce and West [24]. Three grams of frozen tissue powder was homogenized in 10.0 mL 99.5% (*v/v*) ethanol and centrifuged at 20,000× *g* for 20 min. The pellet was dried overnight at room temperature. Fifty milligrams of dried residue was suspended with 5.0 mL of 2 M HCl and 0.5 mL of thioglycolic acid. The sample was heated at 100 °C for 8 h and cooled on ice, then centrifuged at 20,000× *g* for 20 min (4 °C) and the supernate was remove. The pellet was washed with distilled water and re-suspended in 5.0 mL 1 M NaOH. The solution was agitated gently at 25 °C for 18 h and then centrifuged at 20,000× *g* for 20 min. One milliliter of concentrated HCl was added to the supernate and the lignin thioglycolic acid was allowed to precipitate at 4 °C for 4 h. After centrifugation at 20,000× *g* for 20 min, the pellet was dissolved in 1.0 mL of 1 M NaOH. The absorbance was measured against a NaOH blank at 280 nm, and data was expressed on a dry weight basis. Lignin content (mg g^−^^1^ DW) = [(ΔA−0.0068) * V_TA_ * 10^−^^3^ * N]/(0.0347 * m); ‘ΔA’ = A_test_ − A_balnk_; ‘V_TA_’—total volume of the reaction system, mL; ‘m’—dry weight of banana peel, g; ‘N’: ‘0.0068′ and ‘0.0347′ were from the regression equation (y = 0.0347x − 0.0068, R^2^ = 0.9889) under standard conditions.

Superoxide anion and H_2_O_2_ content. Superoxide anion and H_2_O_2_ content were determined using corresponding assay kits (Comin biotechnology Co., Ltd., Suzhou, China) in accordance with the manufacturer’s instructions.

Extration and assay of PPO, POD and PAL. Peel tissues (2.0 g) from three fruit were ground in liquid nitrogen and homogenized in 20.0 mL of 0.05 M phosphate buffer (pH 7.0) and 0.5 g of polyvinylpyrrolidone (insoluble) at 4 °C. After centrifugation for 20 min at 19,000× *g* and 4 °C, the supernate was collected as the crude enzyme extract of the analysis of PPO and POD activity. PAL activity was extracted with 50 mM sodium phosphate buffer (pH 8.8) containing 5 mM β-mercaptoethanol.

According to the method of Jiang [25] with some modifications, PPO activity was measured with 2.9 mL reaction liquid (10 mM pyrocatechol prepared with phosphate buffer) and 0.1 mL enzyme-extracting solution. After adding enzyme-extracting solution, the change in the OD_525nm_ value was measured. One unit of enzyme activity was defined as the amount that cause a change of 0.005 in the absorbance at 525 nm per minute. PPO activity (OD_525nm_ g^−^^1^ min^−^^1^ FW) = ΔOD_525nm_ * Vt/(Vs * m * ΔT * 0.005); ‘Vt’—total volume of enzyme-extracting solution, mL; ‘Vs’—volume of the reactive enzyme liquid added, mL; ‘m’—fresh weight of banana peel, g; ‘ΔT’—reaction time, min.

POD activity was measured with 0.1 mL guaiacol (4.0%), 0.1 mL hydrogen peroxide (0.46%), 2.75 mL phosphate buffer (pH 6.8), and 0.05 mL enzyme-extracting solution. After adding enzyme fluid, the change value of the absorbance of 0.005 at 470 nm was measured. One unit of enzyme activity was defined as the amount that caused a change of 0.01 in the absorbance at 470 nm per minute. PPO activity (OD_470nm_ g^−^^1^ min^−^^1^ FW) = ΔOD_470nm_ * Vt/(Vs * m * ΔT * 0.005).

PAL activity was assayed with the method of Assis et al. [26] with some modifications. A total of 1 mL of the supernate was added to 2.0 mL 50 mM borate buffer (pH 8.8) and 1.0 mL 20 mM 1-phenylalanine and then incubated in a water bath at 37 °C for 2 h. The reaction was stopped by adding 1.0 mL 1 M HCl. Trans-cinnamate formed and the absorbance was assayed at 290 nm. One unit of enzyme activity was defined as the amount that caused a change of 0.05 in the absorbance at 290 nm per minute. PAL activity (OD_290nm_ g^−^^1^ min^−^^1^ FW) = ΔOD_290nm_ * Vt/(Vs * m * ΔT * 0.005). All absorbance values were analyzed by using Multiskan Spectrum and the results were expressed on a fresh weight basis.

Cellulase activity. Cellulase activity was determined by the anthrone colorimetry method. A total of 2 g frozen tissue powder was dissolved in 20.0 mL sodium phosphate buffer (20 mM, pH 7.0) containing cysteine, HCl (20 mM), EDTA (20 mM), and TritonX-100 (0.05%). Then, the homogenate was centrifuged at 15,000× *g* for 30 min at 4 °C and the supernate was collected for enzyme activity analysis. The enzyme was added in sodium acetate buffer (100 mM, pH 5.0) containing carboxy methyl cellulose (1.0% *w/v*) in a final volume of 10.0 mL. The mixture was incubated at 37 °C with vibration for 1 h and was then incubated at 90 °C for 15 min. After cooling down, the mixture was centrifuged at 8000× *g* for 10 min at 25 °C. The supernate was saccharification solution added with authrone and then incubated at 90 °C for 10 min. After another cooling, the samples were analyzed at 620 nm. One unit of cellulase activity is defined as the amount of the enzyme that releases 1.0 μg glucose per milligram of fresh weight per minute. Cellulase activity (μg min^−^^1^ g^−^^1^ FW) = [1000 * (ΔA + 0.0462) * V_TA_ * V_T_]/[2.509 * (m * Vs) * T]; ‘1000′—1 mg mL^−^^1^ = 1000 μg mL^−^^1^; ‘V_TA_’—total volume of the reaction system, mL; ‘Vs’—volume of sample added, mL; ’V_T_’—total volume of enzyme-extracting solution, mL; ’T’—reaction time, min; ‘0.0462’ and ‘2.509’ were from the regression equation (y = 2.509x − 0.0462, R^2^ = 0.9956) under standard conditions.

α-amylase and β-amylase activity. α-amylase and β-amylase were determined with the 3, 5-dinitrosalicylic acid method according to the characteristics that α-amylase was not resistant to acid and β-amylase was not resistant to the heat (70 °C). Assay kits (Comin biotechnology Co., ltd., Suzhou, China) were used to analysis enzyme activity in accordance with the manufacturer’s instructions.

Scanning electron microscopy observation and analysis. The banana peel was cut into regular squares and then fixed with glutaraldehyde (2.5% glutaraldehyde and 2% paraformaldehyde) overnight and vacuumed. The glutaraldehyde was washed with 0.1 M phosphate buffer for 40 min, three times in total, and the samples were then dehydrated with 30%, 50%, and 70% ethanol for 20 min each time, followed by further dehydration with 80%, 90%, and 100% ethanol for 20 min each time. Finally, the samples were washed three times with tert-butanol for 20 min each time. The samples were finally freeze-dried and sprayed with gold (180 s, 20 mA) using the JFC-1600 (JEOL, Tokyo, Japan) ion sputtering instrument. The specimens were observed with a JSM-6360 LV (JEOL, Tokyo, Japan) scanning electron microscope (SEM) at 15 KV.

Quantitative real-time PCR analysis. DNA-free RNA was reverse-transcribed for first-strand cDNA synthesis. The gene-specific oligonucleotide primers were used for qRT-PCR analysis (Appendix A). The qRT-PCR reactions for some genes related with synthesis of lignin were carried out in the ABI 7500 Real-Time PCR System (Applied Biosystems, Carlsbad, CA, USA) with SYBR Green Real-Time PCR Master Mix (TOYOBO Co., Ltd.). The conditions were as follows: 30 s at 95 °C, 40 cycles of 5 s at 95 °C, and 34 s at 58 °C. *MaActin-3* was selected as the reference gene [27]. qRT-PCR reactions were normalized using the Ct value corresponding to that of the reference gene. The relative expression levels of target genes were calculated using the formula 2^−ΔΔCT^. Three independent biological replicates were performed in the analysis.

### 2.3. Statistical Analysis

The experiments were arranged in a completely randomized design with three replicates, and the data were expressed as the mean ± SD (standard deviation). Data were analyzed by SPSS version 19.0. Least significant differences (L.S.D.) were calculated to compare significant effects at the 5% level.

## 3. Results and Discussion

### 3.1. Chilling Symptoms

Many tropical and subtropical fruits are sensitive to low temperatures and suffer different physiological disorders, known as CI. The incidence by CI limits the application of cold storage. Bananas are sensitive to chilling and suffer physiological damage when exposed to low-temperature (<13 °C) storage [18,19,20,21]. As shown in Figure 1A, banana fruit began to exhibit the CI symptoms of pitting and brown patches on the skin after 3 days of storage at 6 °C and became severe as the storage days prolonged when compared with the control group. The hue angle of banana peel decreased significantly on the third day (Figure 1B). the *Fv/Fm* ratio, which refers to chlorophyll content, also decreased as the CI symptoms appeared (Figure 1C), which was consistent with the browning and discoloration symptoms shown in Figure 1A. On the sixth day of storage, the firmness of bananas stored at 25 °C steadily decreased as the storage prolonged and decreased by 11.9% compared with the firmness of bananas stored at 6 °C (Figure 1D). Softening is an important manifestation of banana ripening. The higher firmness might be related to the lignification of peel or abnormal fruit ripening, which indicated that CI might accelerate the lignification of banana peel and affect the normal ripening of banana fruit.

### 3.2. Oxidative Stress, O^2−^ and H_2_O_2_

Alterations in the biomembrane conformation and structure are considered the first events at the molecular level of CI. Different degrees of CI-induced cell membranes have been shown, including the increase in electrolyte leakage [28,29,30], lipid-phase transitions [31], and changes in lipid composition [32]. Among them, low-temperature-induced membrane lipid-phase transition was used to explain the membrane integrity and physiological dysfunction [33]. Apart from the direct effect of low temperatures on the molecular organization of membrane lipids, low-temperature stress boosted the damage of membrane integrity by a disproportionate increase in reactive oxygen species (ROS) production [34]. When the CO_2_ fixation is limited by environment stresses, such as CI, an excess of photosystem I reduction and ROS production is observed [35]. In addition, the activation of NADPH oxidase in cell membrane can induce a massive production of O^2−^ [36]. In our present study, the production rate of O^2−^ in banana peels stored at 6 °C increased during first two days and then decrease in the later storage days (Figure 2A), and it was higher than that of control group by 43.1% on 2nd day. As shown in Figure 2B, for the banana stored at 6 °C, the H_2_O_2_ content of banana peels was higher than that of control group during the entire storage period. It indicated that CI might induce oxidative stress by increasing the content of O^2−^ and H_2_O_2_.

### 3.3. Peel Browning, PPO, POD and PAL

Tissue browning is a common CI symptom in fruit and vegetables [37]. The visual CI symptoms of banana fruit are peel browning and pitting or discoloration. However, an early or mild symptom was darkening of the peel vascular tissues. Such symptoms appear due to enzymatic and non-enzymatic browning reactions involving oxidation of phenolic substrate and pigment degradation [18,38]. Polyphenol oxidase (PPO) and peroxidase (POD) are believed to be a major cause of brown discoloration by the oxidation of phenolic substrates [39] and have synergistic effects on the formation of brown polymers [40]. As shown in Figure 3A, the PPO activity decreased and then increased as storage days prolonged. During the first two days, the banana peel exposed to low temperature (6 °C) exhibited a decrease in the activity of PPO than that of control group. It might be due to the low temperature inhibiting PPO activity at the very onset of storage, while there was dramatically increased PPO activity to response to the cold stress. POD can oxidize phenols to quinones, then condense tannins to brown polymers in the presence of H_2_O_2_, which may contribute to enzymatic browning [41]. Furthermore, POD plays a very important role in the growth and development of plants, such as improving the adaptability of plants to the external environment, increasing the thickness of the secondary wall of plant cells by regulating lignin synthesis, and protecting the cells [42]. As shown in Figure 3B, the POD activity of the two groups showed similar changes that decrease in the first days of storage and then increase in the later storage days. However, the POD activity increased rapidly under low-temperature storage than that of the control group from the fourth day, which might be attributed to the accumulated reactive oxygen species under CI, triggering a rapid increase in POD activity.

Phenylalaninammo-nialyase (PAL) is a vital enzyme between the shikimate pathway and secondary phenylpropanoid pathway, which can convert phenylalanine to mono- and diphenols, which are substrates of PPO [43]. PAL activity could be induced by various stresses, including wounding, UV-B light, ozone, pathogen invasion, and plant hormones [44,45,46]. Parkin et al. [33] suggested that there was an elevation in PAL synthesis activity in fruits occurring at low temperatures, which likely contributed to the increase of phenol concentration in these tissues upon CI. Moreover, the PAL also plays a role in the heat pretreatment-induced chilling tolerance of banana fruit [21]. In our study, for banana stored at 6 °C, PAL activity of banana peel tended to increase with the aggravation of CI symptoms (such as browning) and was significantly higher than that of control group by 70.2% and 92.2% on the 4th and 6th day, respectively (Figure 3C). This was also consistent with the report by Choehom et al. that PAL was involved in the browning reaction [47].

### 3.4. Cell Wall Metabolism, Content of Lignin, Cellulose and Protopectin, Activity of Cellulase and Pectinase

It had been reported that the texture changes of certain CI-sensitive fruits under low-temperature storage were due mainly to disorders of cell wall metabolism, including a decrease in the solubilization and depolymerization of pectin [10].

Lignin is the major constituent of the secondary cell wall in plants and is involved in plant growth, development, and defense [48]. In our study, the lignin content of banana peels stored at 6 °C was increased from 30.03 to 35.04 mg g^−1^ DW, while the lignin content of banana peels stored at 25 °C did not change significantly (Figure 4A). The increase of lignin content was consistent with the increase in firmness, which indicated that the low temperature promoted the lignification of banana fruit. Cellulose is the most dominant constituent of plant cell walls [49]. As shown in Figure 4B, the cellulose content of banana stored at 25 °C decreased as the storage days prolonged, while the bananas stored at 6 °C exhibited a non-significant change. It indicated that CI inhibited the degradation of cellulose and then affected cell wall metabolism. Pectin is a common component of the middle lamella and the primary cell walls in the fruit texture, constituting approximately one third of the structure [50]. Pectin solubilization is a common feature of fleshy fruit ripening. As shown in Figure 4C, the protopectin content of banana peels stored at 6 °C increased rapidly and then decreased slowly, which showed a higher level than the control group. The changes of fruit texture are closely related to the degradation of cell wall components, such as the degradation of protopectin and the increase of soluble pectin content [51]. In our study, low temperature could cause metabolic disorder of the cell wall and inhibit the degradation of cellulose and protopectin, thus inhibiting the post-ripening of banana fruit and promoting lignification of banana peels.

The major textural changes resulting in the fruit softening are related to enzyme-mediated alterations in the structure and composition of the cell wall, including the changes in cell wall composition mentioned above, as well as the changes of related enzyme activity. As shown in Figure 5A, in the first two storage days, the cellulase activity increased gradually, which was consistent with the decrease in cellulose content for the two groups. As the storage period was prolonged, the cellulase activity decreased for bananas stored at 25 °C, and the cellulose content increased form the forth storage day. However, for bananas stored at 6 °C, the cellulase activity for banana peels basically remained unchanged, while cellulose content increased slightly (Figure 5A). On the fourth and sixth day of low-temperature storage, the cellulose content was 53.79% and 25.43% higher than that stored at 25 °C, respectively (Figure 4B), which indicated that low-temperature storage affects the cell wall component. As shown in Figure 5B, for the bananas stored at 25 °C, pectinase activity was decreased during the whole storage period, while the protopectin content was also decreased, which might due to protopectin being not the only substrate of pectinase, which needs further investigation. For the pectinase activity of banana peel at 6 °C, it rapidly decreased during the first storage days and then increased in the later storage days, and the protopectin content was increased first and then decreased (Figure 4C and Figure 5B). In the early storage stage, pectinase activity was inhibited by cold stress, and with the enhancement of adaptability to low temperature in the later stage, the pectinase activity was increased to degrade the content of pectin. These results indicated that chilling injury caused the disorder of metabolism of the banana peel cell wall and finally led to the change in the cell wall composition.

As the results above, bananas could change the structure of their cell wall by affecting cell wall metabolism to respond to cold stress during storage at a low temperature.

### 3.5. Ultrastructural Changes of Banana Peel after Chilling Injury

In addition to the structural alteration and rearrangements of the cell wall, the tissue structure was also significantly changed by scanning electron microscopy (SEM) observation. As shown in Figure 6A, the boundary of epidermal cells on the banana peels stored at a low temperature were blurred. Starch is the main form of carbon storage in bananas, and the unripe bananas have a large amount of starch. In our present study, the starch grains in banana peels stored at 25 °C decreased gradually as the storage time prolonged and were almost degraded on the 6th day of storage, while the starch grains in banana peels stored at 6 °C basically remained unchanged (Figure 6B). Furthermore, the α/β amylase activity of banana peels were analyzed. During the storage period, the α-amylase content in banana peel of the two treatment groups showed a trend of rapid decrease in the first two days and then remained basically unchanged (Appendix A). The β-amylase in banana peels remained basically unchanged under low-temperature storage, while it increased significantly in the control group (Appendix A). This was consistent with the decrease of starch grains in Figure 6B. These results indicated that β-amylase might play a vital role in the degradation of starch grains in banana peels, and CI might inhibit the post-ripening of bananas by affecting the degradation of starch grains. In addition, we found that the tissue structure and cell structure (such as vascular bundle cells) of the longitudinal section of banana peels had no significant changes before and after CI (Appendix A).

### 3.6. Expression Profiles of Many Genes Involved in Lignification Caused by CI

To further understand the mechanism of lignification in bananas after CI, some genes possibly involved in the lignification process were selected for gene expression analysis, as shown in Figure 7, including *POD1*, *PAL*, *LAC3*, *CCR4*, *CAD2,* and *4CL7*. Studies have shown that POD and PAL are the key enzymes of lignification, and the enzyme activities increase significantly as lignification occurs [52,53]. POD is an oxido-reductive enzyme that participates in the cell wall polysaccharide processes, such as the oxidation of phenols, suberization, and lignification [42]. In our present study, the gene expression of *POD1* for chilled banana peels was down-regulated at the early storage stage, then up-regulated from the second day, and finally higher than that of the control group at the end of the storage period (Figure 7A). This is consistent with the change of POD activity (Figure 3B). LAC has also been reported to be implicated in the polymerization of lignin and is required for lignification in plants [54,55]. It has been reported that *AtmiR397b* can reduce the lignin deposition by negatively regulating *LAC4* [56]. *ZmmiR528* regulates lignin biosynthesis and lignification by negatively regulating *LAC3* and *LAC5* [57]. In our study, the gene expression of *LAC3* increased dramatically, especially on the fourth day (Figure 7C), which was consistent with the change in PAL activity. It indicated that *PAL*, *POD,* and *LAC* play important roles in the lignification of banana peels under chilling injury.

Lignin is one of the important products of the phenylpropane metabolism pathway, and its biosynthesis involves many enzymes, such as CCR, CAD, and 4CL. CCR is the first rate-limiting enzyme in the lignin synthesis pathway, which has a potential regulatory effect on lignin monomer biosynthesis [58]. CAD is the earliest studied enzyme in the lignin synthesis pathway and is the last step to catalyze the formation of the lignin monomer [59]. CAD can convert aldehydes into alcohols and plays an important role in the proportion regulation of lignin monomer synthesis [60,61,62]. At the terminal position is 4CL in the phenylpropanoid derivative metabolic pathway and is the last enzyme that turns the phenylpropanoid metabolic pathway to the downstream branch pathway. As the key enzyme that connects the phenylpropanoid metabolic pathway to the specific lignin synthesis pathway, 4CL plays a rate-limiting role in the lignin monomer synthesis pathway [63]. In our study, the gene expressions of *CCR4*, *CAD2,* and *4CL7* were significantly up-regulated during the storage for chilled banana fruit, which was involved in the lignification process for bananas after chilling injury (Figure 7D–F). It indicated that under cold stress, the lignin synthesis-related genes were up-regulated to promote the lignification of banana peels, which could reinforce the cell walls to alleviate the damage caused by cold stress.

## 4. Conclusions

Low-temperature storage can improve the storage quality of fruits and vegetables but also has adverse effects on heat sensitive fruits and vegetables. The study comprehensively elucidated the involvement of chilling symptoms and the mechanism of lignification for bananas after chilling injury. In the present study, banana peel browning induced by chilling injury was promoted by the degradation of chlorophyll and the increased activity of POD and PAL. Chilling injury accelerated the senescence of banana fruit by increasing the content of O^2−^ and hydrogen peroxide. In addition, chilling injury affected the normal post-harvest ripening process of banana fruit by inhibiting the normal degradation of starch grains and metabolism of the cell wall. We found that lignification, as a response to abiotic stress and senescence, is an important manifestation of banana fruits after chilling injury. The study showed that *PAL* might start the phenylpropanoid pathway of lignin synthesis, while the synthesis of lignin monomers was promoted by up-regulating the expression of the *CCR4*, *CAD2*, and *4CL7* genes. Then, the *POD1* and *LAC3* genes were up-regulated to promote the oxidation of lignin monomers. *POD1* and *LAC3* were finally up-regulated to promote the oxidative polymerization of lignin monomers. These results suggest that changes in cell wall structure and metabolism, as well as lignification, are involved in the senescence and quality deterioration of bananas after chilling injury. Studies on the characteristics and lignification mechanisms of bananas involved in chilling injury will provide a theoretical basis for heat-sensitive fruits and vegetables to alleviate CI.

## Figures and Tables

**Figure 1 foods-12-01097-f001:**
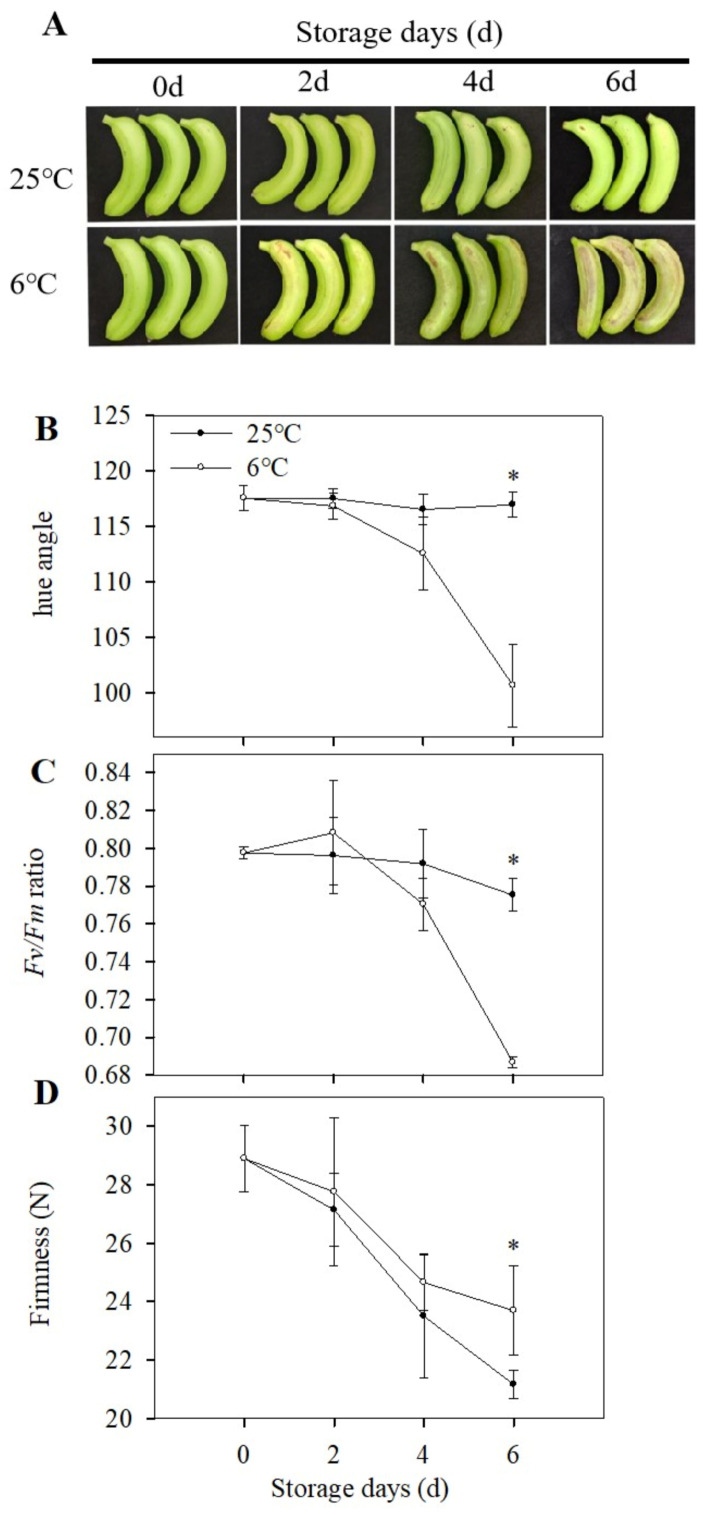
Chilling injury characteristics of banana fruit during storage at 6 °C. Fruit stored at 25 °C is the control. Changes in chilling injury phenotype (**A**), hue angle (**B**), *Fv/Fm* (**C**), and firmness (**D**) of banana fruit during storage. Each data point represents a mean ± standard error (*n* = 3). ‘*’ means a significant difference between control and experimental fruit at the 5% level.

**Figure 2 foods-12-01097-f002:**
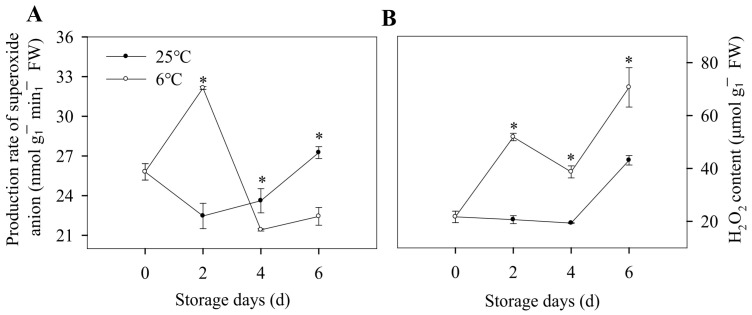
Changes in oxidative stress of banana fruit during storage at 6 °C and 25 °C. Production rate of superoxide anion (**A**) and H_2_O_2_ content (**B**) of banana peels during storage. Each data point represents a mean ± standard error (*n* = 3). ‘*’ means a significant difference between control and experimental fruit at the 5% level.

**Figure 3 foods-12-01097-f003:**
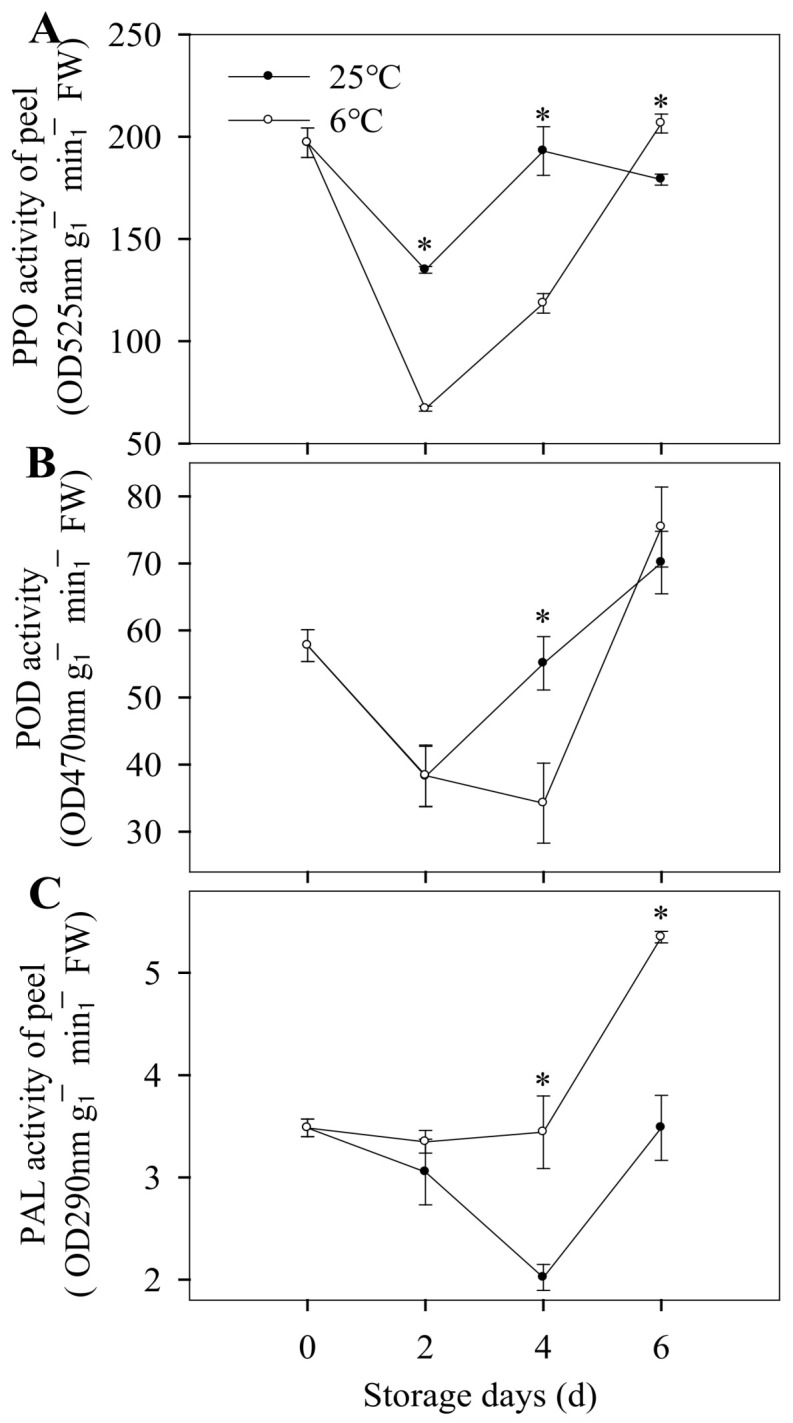
Changes in enzyme activity associated with browning of banana fruit during storage at 6 °C and 25 °C. PPO activity (**A**), POD activity (**B**), and PAL activity (**C**) of banana peel during storage. Each data point represents a mean ± standard error (*n* = 3). ‘*’ means a significant difference between control and experimental fruit at the 5% level.

**Figure 4 foods-12-01097-f004:**
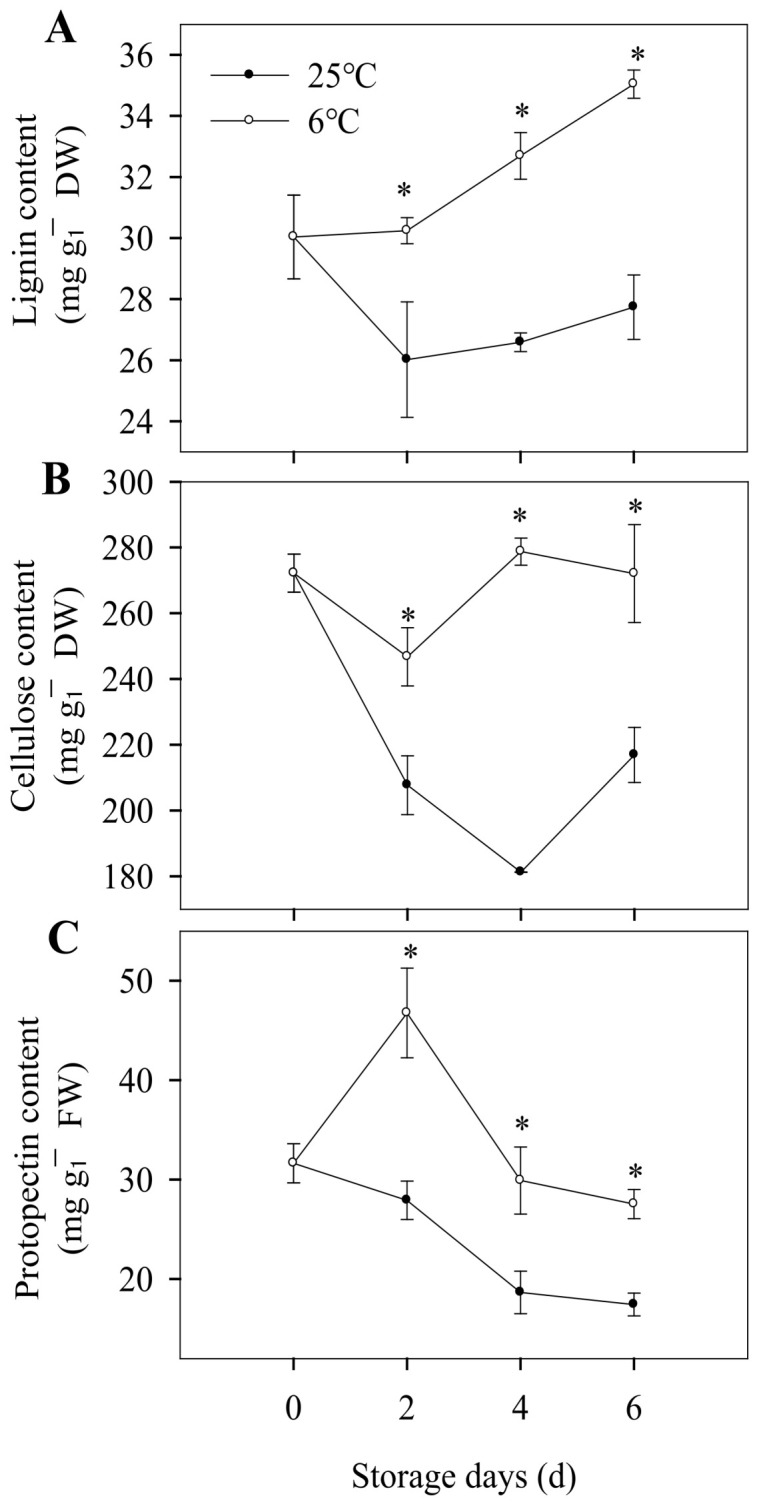
Changes in cell wall component of banana fruit during storage at 6 °C and 25 °C. Lignin content (**A**), cellulose content (**B**), and protopectin content (**C**) of banana peels during storage. Each data point represents a mean ± standard error (*n* = 3). ‘*’ means a significant difference between control and experimental fruit at 5% level.

**Figure 5 foods-12-01097-f005:**
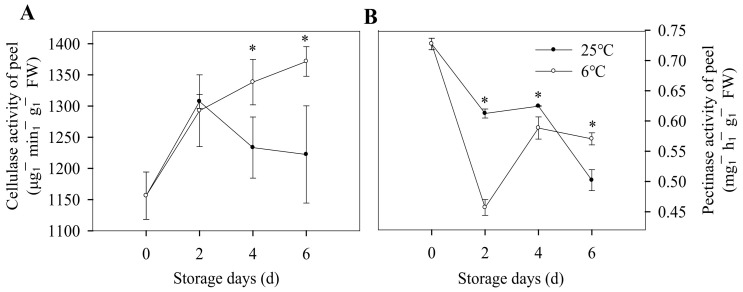
Changes of enzyme activity related to cell wall degradation of banana fruit during storage at 6 °C and 25 °C. Cellulase activity (**A**) and pectinase activity (**B**) of banana peels during storage. Each data point represents a mean ± standard error (*n* = 3). ‘*’ means a significant difference between control and experimental fruit at 5% level.

**Figure 6 foods-12-01097-f006:**
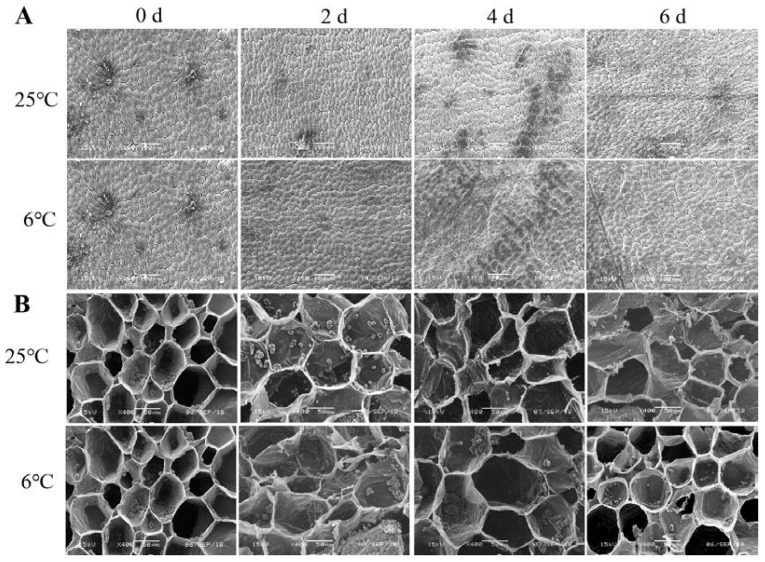
Structural changes of banana peel were observed by scanning electron microscope during storage at 6 °C and 25 °C. Epidermis structure (**A**) and Starch granules (**B**) of banana peel during storage.

**Figure 7 foods-12-01097-f007:**
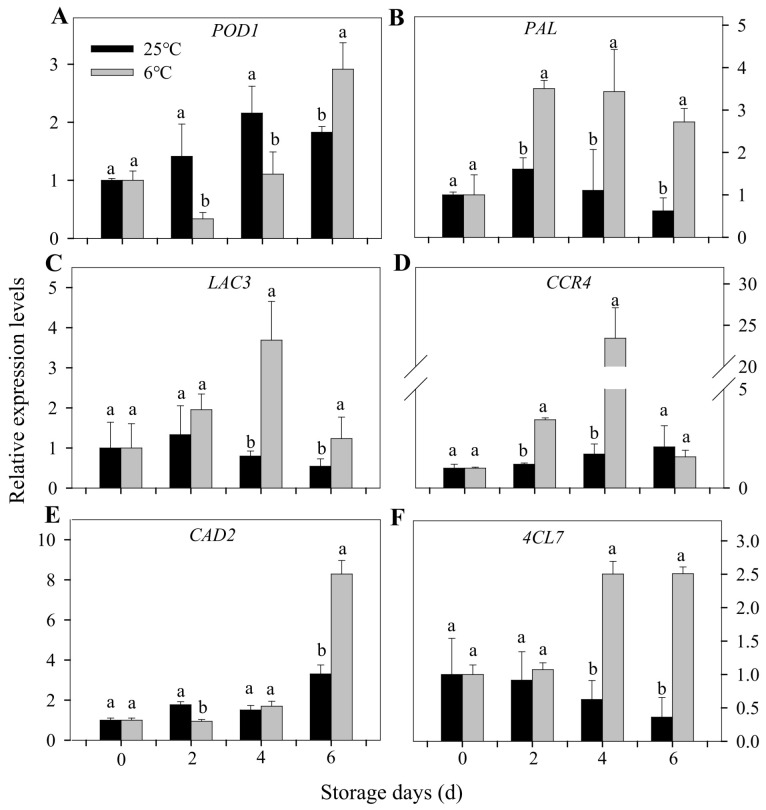
Changes in the relative expression of genes associated with lignification in banana peels during storage at 6 °C and 25 °C. *POD1* (**A**), *PAL* (**B**), *LAC3* (**C**), *CCR4* (**D**), *CAD2* (**E**), and *4CL7* (**F**). Each data point represents a mean ± standard error (*n* = 3). The values with different letters for different storage temperature for each sampling date are significantly different (*p* < 0.05).

## Data Availability

Data is contained within the article or Appendix A.

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
