# Peer review of "Study on Characteristics and Lignification Mechanism of Postharvest Banana Fruit during Chilling Injury"

_foods, 2023, doi:10.3390/foods12051097_

Round 1

Reviewer 1 Report

The authors investigated lignification mechanism, as a response to abiotic stress and senescence of banana fruits after Chilling injury. The work is interesting and highlight some novel findings. There are some suggestions that needs to be considered.

Abstract: Please move L16 to L21 to Introduction and add some methodology and some results (values, % difference etc.).

Use full terms for enzymes in the Abstract.

Keywords: Use keywords that are not part of the title.

Introduction:

Chilling injury is well known physiological disorder. Therefore, I would recommend to shorten paragraph L40-57, and discuss how chilling injury cause lignification. Lignification is the major focus of this study but it is confined to only L58-61. So, mechanism of lignification as a function of chilling injury may be a bit more discussed.

Materials and Methods:

Please add coordinates for the orchard.

Explain terms in full before being abbreviated.

Cinnamyl alcohol dehydrogenase is one of the most important lignification enzymes. It is better to measure the activity of this enzyme.

Results and Discussion:

Results are poorly presented and wordy. This section can be more interesting and understandable if numerical values or comparisons (% difference etc.) are given.

What is the reason that cellulose content was higher at 6 C, whereas Cellulase activity was also higher at 6 C. Logically, there should be antagonistic relationship between enzyme activity and substrate, which is exact the opposite in this case.

Same is the case with pectin and pectinase enzyme. It needs explanation in the light of sound evidence.

Conclusion:

Delete L398-403. Conclusion lacks any conclusion.

At some places, language needs improvement as L35-36, L70, L75 etc.

Author Response

We thank you so much for your positive comments and providing us the opportunity to revise this manuscript. Those comments are valuable and very helpful. According to Editor’s and reviewer’s comments, we have read through comments carefully and tried our best to revise the manuscript. Based on the instructions provided in your letter, we amended the relevant part and uploaded the file of the revised manuscript. Relevant revisions have been marked in red color.

Reviewer 2 Report

The manuscript needs some major corrections.
Points for the authors to address:

-writing should be correct and precise

-some analyses are not presented appropriately (please find the details bellow)

-results are not explained clearly enough and compared with similar research

Specific comments

Line 92

Firmness check model 327 - for bananas?

Line 94-95

Units in Results?

Line 96

Peel color:

-Color results - which parameters were measured?

-What is H value in Figure 1 (B)? Is it hue angle (h°)?

Line 148

According to...?

Line 150

Enzyme liquid?! Enzyme manufacturer?

Line 257

During the first three?...

Figure 3 (B)

Results for control sample-day 2 missing!

Figures S1 and S2

The same caption!

Suggestion for future research determination of pectic substances, not only protopectin content.

Author Response

We thank you so much for your review and providing us the opportunity to revise this manuscript. Those comments are valuable and very helpful. According to Editor’s and reviewer’s comments, we have read through comments carefully and tried our best to revise the manuscript. Based on the instructions provided in your letter, we amended the relevant part and uploaded the file of the revised manuscript. Relevant revisions have been marked in red color.

Reviewer 3 Report

See attached file

Author Response

(The authors gave the same response as above.)

Reviewer 4 Report

Dear Editor The authors has planned very well this experiment. This study reveals interesting information for readers. There are several mistakes which need to be corrected before the final acceptance. Abstract is very generally and doesn't show the significance of the study, authors need to rewrite the result parts and must describe how the experiment carried out. In the introduction part, third paragraph shows general information about lignificaiton. So, authors need to specifically give more description regarding the previous studies on the lignification process or main their finding in the introduction. Material and method part is relatively weak which require more attention, also mention the stage of banana harvesting, authors didnt mention the how authors measured protopectin content, formula, instrument etc..also must mention the what kind of instrument used for the taking absorbance? Similarly, in the extraction and assay of PPO, POD and PAL,and authors should need to mention their units. authors must mention the formula for the measurement of PPO and PAL and Cellulose activities.

Author Response

(The authors gave the same response as above.)

Round 2

Author Response

Deer editor and reviewers:

    Thanks for your email informing us that our manuscript (Manuscript ID: foods-2131199 ) was due for revision again. We appreciate that you gave us a chance of minor revision to improve our manuscript. According to Editor’s and reviewer’s comments, we have read through comments carefully and made responding revise in the manuscript, relevant revisions have been marked in red color. Based on the instructions provided in your letter, we uploaded the file of the revised manuscript.

Kind regards,

Lu Xiao

Reviewer 3 Report

The current manuscript is a revised version of a previous submission. Authors have followed most of the comments and suggestions of the revision made comprehensive changes that I think improved the quality of the manuscript. I suggest revision of the figure legends, indicating the two-storage temperature. In summary, I recommend acceptance of the manuscript but I suggest some editing of the English version.

Author Response

(The authors gave the same response as above.)

Reviewer 4 Report

Dear Editor

Thanks. I have thoroughly reviewed the revised version of the paper. Authors have effectively revised the manuscript and addressed all points.

Author Response

(The authors gave the same response as above.)
